# Non-Contact VITAL Signs Monitoring of a Patient Lying on Surgical Bed Using Beamforming FMCW Radar

**DOI:** 10.3390/s22218167

**Published:** 2022-10-25

**Authors:** Sungmook Lim, Gwang Soo Jang, Wonyoung Song, Baek-hyun Kim, Dong Hyun Kim

**Affiliations:** 1AU Inc., Daejeon 34139, Korea; 2SMG-SNU Boramae Medical Center, 20, Boramae-ro 5-gil, Dongjak-gu, Seoul 07061, Korea

**Keywords:** FMCW, radar, vital signs, heartbeat, respiration, beamforming, interferer

## Abstract

Respiration and heartrates are important information for surgery. When the vital signs of the patient lying prone are monitored using radar installed on the back of the surgical bed, the surgeon’s movements reduce the accuracy of these monitored vital signs. This study proposes a method for enhancing the monitored vital sign accuracies of a patient lying on a surgical bed using a 60 GHz frequency modulated continuous wave (FMCW) radar system with beamforming. The vital sign accuracies were enhanced by applying a fast Fourier transform (FFT) for range and beamforming which suppress the noise generated at different ranges and angles from the patient’s position. The experiment was performed for a patient lying on a surgical bed with or without surgeon. Comparing a continuous-wave (CW) Doppler radar, the FMCW radar with beamforming improved almost 22 dB of signal-to-interference and noise ratio (SINR) for vital signals. More than 90% accuracy of monitoring respiration and heartrates was achieved even though the surgeon was located next to the patient as an interferer. It was analyzed using a proposed vital signal model included in the radar IF equation.

## 1. Introduction

Respiration and heartrates are important information for surgery. The traditional methods for monitoring human vital signs, such as electrocardiogram (ECG) and pulse oximetry, need to contact sensors on the body’s surface with bulky monitoring machines and lines should following nearby. This can limit the surgeon’s movements. Therefore, in surgery, the vital signs should be monitored without contact. To monitor the vital signs without contact, a radar sensor has been considered in much research [1,2,3,4,5,6,7,8,9]. However, when the surgeon performs surgery for the patient lying on the surgical bed, the surgeon’s movements reduce the vital sign accuracies monitored by the radar. 

Thanks to the demand for sensing remotely, the radar sensor has received attention and started entering the industry market. In the healthcare industry, using the technologies based on monitoring vital signs and movements, radar sensors have been applied to fall detection [10,11,12], vital sign monitoring, and sleep monitoring [13,14,15,16,17]. As well, beamforming imaging radar is applied to cancer diagnoses [18,19,20]. In the automotive industry, the radar function of monitoring respiration and heartbeat applies to occupant detection [21,22,23,24,25] and driver monitoring system [26,27,28]. Some companies already sell cars with occupancy detection radar. In disaster situations, radar is applied to detect peoples’ conditions [29,30,31].

Most studies on radar for non-contact monitoring vital signs have two main characteristics. The first characteristic is the type of radar. In the beginning, the CW Doppler radar was used [1,2,3,4]. In CW Doppler radar, the vital signs are extracted from the phase difference between the transmitted and received waveforms, which is caused by chest movements due to breathing and heartbeat. The CW Doppler radar system is very simple and relatively cheap; however, it cannot distinguish between two or more objects at different ranges simultaneously. Some recent studies have adopted the ultra-wide-band (UWB) [5,6,7,8,9] or FMCW radar [15,32,33,34,35,36,37], which can distinguish several objects at different ranges.

The second characteristic is that the frequency band of radar has been higher. With low-frequency bands, the radar system can be implemented easily with commercial sub-block chips but it becomes very bulky with large antennas and matching networks. In addition, the bandwidth is limited to lower frequencies and causes low resolution and reduced accuracy of vital sign monitoring data. Thanks to the remarkable development of the IC technology, millimeter-wave bands such as 24 and 60 GHz can be used for radar. Recent studies reported the 60 GHz radar monitoring vital signs [2,36,37,38]. At these frequencies, the radar system can be integrated into a single chip with a very small area. The small form factor can array multiple antenna in a limited area. The antenna array makes it possible to use beamforming which can achieve high-angle resolution.

When considering the field of application, the attached radar’s position is an important variable. Some studies have shown that the radar system can be attached to the ceiling and it can monitor the respiration and heartrates of a patient lying on a bed [39,40]. In such a case, the vital sign accuracy can be reduced when there are people or objects moving around the patient. In our application, if radar monitoring vital signs is used in the operating room, installing the radar on the ceiling would make it impossible to monitor the vital signs of the patient due to the surgeon’s body and arm movements. Therefore, attaching radar to the ceiling is not suitable for our application. To solve the issue of interferer between radar and patient, the radar should be installed on the back of the bed. 

In this study, when the radar was installed on the back of a bed, a method for enhancing the accuracies of monitored respiration and heartrates the for patient was proposed. Installing radar on the back of the bed made additional noise which was generated by electromagnetic-wave reflections from the bed structures. Almost all previous works use radar which was set at the front of the target. It was not affected by reflections because the space between radar and target was opened. The FMCW and beamforming were used to suppress the noise of different angles and ranges from the patient’s position. As a result, the monitored respiration and heartrate accuracies were enhanced even though the surgeon was next to the patient. To analyze the accuracy improvement by FMCW and beamforming, the radar IF signal model including respiration and heartbeat signal was proposed.

The experiment was performed using the 60 GHz FMCW radar module based on TI’s AWR6843AOP chip with 3-TX and 4-RX antennas on the package, which is shown in Figure 1a. The radar was installed on the back of the bed and both the respiration and heartratesof the patient lying on the bed were measured, as shown in Figure 1b,c. The beamwidth of the patch antenna in our radar was 120 degrees. The angular resolution was 29 degrees using 4 × 4 MIMO architecture, which is shown in Figure 2a.

## 2. Algorithm Monitoring Respiration and Heart Rates

In our 3-TXs and 4-RXs radar system, the ADC output raw data was generated for virtual 12-channels whose architecture is shown in Figure 2a. As is shown in Figure 2b, the raw data was processed and re-organized to a 4-dimensional cube which was composed of chirp, range, azimuth, and elevation angles. 

Figure 3 shows the flow chart of the signal-processing algorithm extracting vital signs from the raw data. First, the FFT was applied to ADC samples in order to calculate the range of the target. Thereafter, digital beamforming was applied to 12-channel signals of the azimuth and elevation direction using Bartlett’s technique. Subsequently, a range-azimuth-elevation radar cube was obtained for each chirp. To shrink from a 4D cube to a 1D chirp signal, complex value data of the range and angles matched to the target’s position was accumulated over 640 chirps which corresponded to 20 s. A low-pass filter (LPF) with a finite impulse response (FIR) filter was applied to the accumulated signal to remove high-frequency noise and the phase was extracted from all complex values. The cut-off frequency of the FIR filter was 30 Hz because the heartbeat had the high-frequency component even though the heartrate was under 4 Hz (=240 BPM). Another LPF with a moving average was applied to the phase signal. The respiration waveform was extracted, removing static clutter based on the DC component from the result. The heartbeat waveform was obtained by subtracting the respiration waveform before removing static clutter. The subtraction can remove static clutter for the heartbeat. Since the heartbeat is not a general sinusoidal wave, an additional LPF was applied to the heartbeat waveform. Finally, FFT was applied to the respiration and heartbeat waveforms to obtain the respiration and heartrates, respectively. 

## 3. Study on the SINR Improvement by FMCW Radar with Beamforming

When a surgeon operates on a patient, the surgeon occupies several range and angle bins in the aspect of radar. Therefore, the noise caused by the surgeon’s movements reduces the SINR of the patient’s respiration and heartbeat signals. The decrease in SINR makes it difficult to find respiration and heartbeat from the phase signal and reduces the accuracy. In order to suppress the decrease in SINR, signal processing should be performed on the phase signal matched for the patient’s range and angle. It means that the radar should separate the range and angle.

We modeled respiration and heartbeat signals to confirm the increase in SINR for vital signals by applying range-FFT and beamforming. Since the respiration and heartbeat signal model was based on the radar IF signal, it can be obtained in the same format as the ADC output raw data received from the radar.

Through the simulation using the proposed model, the SINR improvement effect by range-FFT and beamforming was confirmed, respectively.

### 3.1. Modeling a Radar If Signal including Vital Signals

A vital signal model based on radar IF signal equation was devised. As shown in Equation (1), the radar IF signal model is based on a complex exponential function. It represents a sum of five terms in phase. The first term represents the frequency of the ADC sample points related to the range information. The second and third terms represent the phase variation across chirps caused by the movements due to respiration and heartbeat. The fourth and fifth terms express the phase difference among the Rx channels that was caused by the different arrival angles of each Rx channel.
(1)$$S_{IF}=A\exp\left\{2\pi ft+\phi_{RP}+\phi_{HB}+\frac{2\pi }{\lambda }d_{Az}sin\theta_{Az}\;\left(N_{Az}-1\right)+\frac{2\pi }{\lambda }d_{El}sin\theta_{El}\;\left(N_{El}-1\right)\right\}$$

Movement due to respiration is a sinusoidal wave in the phase signal, which means periodic movement back and forth in a specific path in a complex plane. This can be expressed as Equation (2). The length of the round-trip path is given by the amplitude $A_{RP}$ of phase variation and respiration rate is the frequency $f_{RP}$; since the phase variation is in chirps, the time term is chirp duration time $T_{c}$.
(2)$$\phi_{RP}=A_{RP}\cdot \cos\left(2\pi f_{RP}T_{c}\right)$$

The heartbeat should be modeled using different type of base function, since the feature of heart movement is different from chest movement by respiration. Unlike the case of respiration, heartbeat cannot be modeled by a general sinusoidal function. Heartbeat signal is a wavelet function and is repeated with a heartbeat rate. Heartbeat can be modeled with the convolution of a wavelet function and an impulse train, as expressed in Equation (3). $A_{HB}$ is the amplitude of heartbeat and $\tau_{Heartbeat\;period}$ is the heartbeat rate.
(3)$$\phi_{HB}=\overset{\infty }{\underset{k=0}{\sum }}A_{HB}\cdot f_{wavelet}\left(T_{c}\right)*\delta \left(T_{c}-k\tau_{Heart\;beat\;period}\right)\;$$

The waveforms of proposed respiration and heartbeat models are shown in Figure 4a,b respectively. Inserting Equations (2) and (3) in Equation (1), the final radar IF equation could be obtained, considering both respiration and heartbeat signals.

### 3.2. Simulation with the Proposed Radar IF Signal Model

Using the proposed model radar IF model, the simulation was performed for the virtual target whose respiration rate was 20 beats per minute (BPM) and heartrate was 70 BPM. Figure 5 shows the simulation results. Figure 5a shows the chirp-to-chirp phase variation signal of 1-RX channel in complex plane. The trace of the round trip was related to movements of respiration and heartbeat. The round trip signal can express in-time domain, separating real and imaginary parts. Figure 5b shows the real and imaginary part signals for 20 s; the black dashed line is real part signal and the blue solid line is imaginary part signal. The obtained phase signal from the real and imaginary part signal is shown in Figure 5c. The phase signal contained both respiration and heartbeat signals and matched real measured data which is shown in Section 5.

When extracting respiration waveforms, it is proper to use the phase signal. Assume that the quadrant was changed during a period in complex domain like Figure 5a, the frequencies of real and imaginary signals was different; it was determined by the phase information from radar to target. Occasionally, it is difficult to know whether the respiration signal is a real or imaginary part.

### 3.3. Simulation with the Interferer in a Different Range

To confirm the SINR improvement effect by range-FFT, the phase signals were compared with and without an interferer at a different distance from the target. For the case without an interferer, the proposed model in Section 3.2 was used. For the case with an interferer, the radar IF signal was generated by summing up the two radar IF signals for the target and interferer based on the information in Table 1. The simulation was performed by adding noise to the signal model. The chirp parameters for the radar IF signal model were the same as values of our radar. The chirp bandwidth was 750 MHz and the chirp slope was 24. The range resolution was 20 cm. Considering the maximum required range, 64 samples in the time domain was used.

Figure 6a shows the ranges of target and interferer. They were obtained by applying range-FFT to the fast time signal. 

Without applying range-FFT, when data was processed like CW Doppler radar, the phase signals obtained with and without an interferer are shown in Figure 6b. The noise added to the signal model appeared in the phase signal as it is. In particular, for the case with an interferer, the phase signal is distorted by the noise of the interferer’s movements which had a different frequency from the target’s respiration.

Figure 6c shows the phase signal obtained by applying range-FFT. Through range-FFT, signal distortion caused by the movements of the interferer and noise from other ranges was suppressed. The SINR was improved by 9.08 dB by applying range-FFT. The improvement can be seen by comparing the signals of blue lines in Figure 6b,c. As a result, the phase signal is almost similar to the cases with and without the interferer.

### 3.4. Simulation with the Interferer in a Different Angle

To confirm the SINR improvement effect by beamforming, the phase signals were compared with and without an interferer at a different angle but within the same range from the target. To use beamforming, radar IF signals for the 12-RX channel were generated with noise. To match the simulation with the experiment, the antenna architecture of 12-RX was kept the same as that of AWR6843AOP with an L-shaped TX antenna array and 2 × 2 RX antenna array. When applying beamforming, the number of angle bins was 32 for each axis. Information regarding both target and interferer is listed in Table 2. Signals of the target and interferer were eventually generated and summed.

The target’s range and angle position were extracted by applying range-FFT along with beamforming. Figure 7 shows a heat map of the calculated results. The position of the target and interferer can be seen in the color of the heat map, indicating the signal power. 

Figure 8a is the phase signals to which range-FFT is applied. Because range-FFT cannot suppress noise from different angles, signal distortion occurred for the case with interferer at different angle.

The phase signal obtained by applying beamforming is shown in Figure 8b. It can be confirmed that noise from other angles was suppressed through beamforming. There is no distortion in the phase signal even if there is an interferer at a different angle and the phase signal was almost similar to the case without an interferer. As a result, SINR was additionally improved by 6.38 dB. Totally, 14.58 dB of SINR was improved through range-FFT and beamforming.

Because the surgeon as interferer occupies several range and angle bins, the range-FFT and beamforming should be applied together.

## 4. Experimental Setup

Hardware of the proposed radar system is shown in Figure 2a. It was implemented on the basis of TI’s AWR6843AOP chip, which operates at 60 GHz band (60–64 GHz). There were 3-TX and 4-RX channels with packaged patch antennas. The TX antennas were arranged in an “L” shape and the RX antennas were arranged in a rectangle shape as shown in Figure 2b. The multi-input multi-output (MIMO) virtual antenna array is shown in Figure 2c. There were four virtual channels in the azimuth and elevation directions, simultaneously. The radar was mounted on the backside of the bed as illustrated in Figure 9a. The radar’s 12 dB TX power and 30 dB RX gain were sufficient to transmit an electromagnetic wave to penetrate surgical bed.

The measurement setup is shown in Figure 9a,b. Two types of situations were assumed to prove the improvement by the proposed vital monitoring radar system. The first was a patient lying on the bed alone, as shown in Figure 9b. The second was a patient lying on the bed with a surgeon moving around. It can be shown in Figure 9c. The patient was wearing reference sensors to compare with the radar’s processed results. The Zephyr’s Bio Harness was used as a reference of heartbeat and respiration rate, respectively.

Signal processing was embedded in the microcontroller unit (MCU) and the digital signal processor (DSP), which were in turn integrated in the TI’s AWR6843AOP chip. The final respiration and heartbeat waveforms and rates were transmitted from the radar hardware to a laptop in real time. The processed results were transmitted every second.

## 5. Results and Analysis

Using the proposed algorithm, we could extract both respiration and heartbeat waveforms. The phase signal applying range-FFT and beamforming was filtered using a moving average type filter and the respiration waveform was obtained, as shown in Figure 10a. The heartbeat waveform was extracted by subtracting the original phase signal from the respiration waveform; the results are presented in Figure 10b. 

Both waveforms were almost matched to the proposed signal model. The extracted heartbeat waveform was similar to the seismocardiogram (SCG) which is related to mechanical movement of heart [41,42]. 

### 5.1. Analysis the Results of Case without Interferer

Results showed that the application of range-FFT and beamforming to the radar output signal improved both SNR and accuracy in the monitoring of vital signals. The results are shown in Figure 11a–d. Figure 11a is the phase signal when operating like a CW Doppler radar. The range-FFT and beamforming were not applied in this case. It is difficult to recognize respiration and heartbeat from the phase signal due to low SNR. In our application, a radar was installed on the back of the bed. Therefore, the radar interacts with the stainless steel frame and mattress; multiple reflections can occur and it generates the noise in the radar IF signal and reduces the SNR.

Applying range-FFT could reduce the noise in the phase signal; then, we could recognize the respiration waveform from the phase signal. This is shown in Figure 11b. However, it is difficult to find the heartbeat from the phase signal due to insufficient SNR. 

The multiple reflections could create the noise from different angles than the target. The noise cannot be reduced by applying range-FFT. To remove the noise of different angles, the beamforming should be applied. 

Beamforming was applied to the range-FFT result of all RX channels and the vital signal occurred at 0° of azimuth and 0° of elevation. As shown in Figure 11c, the noise was dramatically reduced. In time domain, the improvement can be seen by comparing the signals in Figure 11a,c. Applying FFT to the signals in Figure 11a,c, we calculated the amount of improved SNR. The amount of improved SNR was almost 20 dB with applying range-FFT and BF. Then, we could recognize the heartbeat in the phase signal. Figure 11d shows the heartbeat waveform extracted from Figure 11c using our algorithm. The measured respiration and heartrates for 100-s were drawn in Figure 12. When there wasn’t an interferer, the algorithm performances of the three cases (raw phase signal, applying range-FFT, and applying range-FFT and beamforming) were compared. Only in the case of applying range-FFT and beamforming were the respiration and heartrates sustained. For the case of applying range-FFT, the fluctuation of heartrate was caused by noise from multiple reflections.

### 5.2. Analysis of the Results of the Case with an Interferer

The presence of an interferer next to the target can distort the phase signal, except for random noise. It has affects at a different range and angle because the interferer is a moving human body. This is shown in Figure 13a–d. Figure 13b shows the phase signal after applying range-FFT. The phase signal is not similar to a sinusoidal wave ignoring the noise. The distorted phase signal from 2 s to 8 s is similar to simulated results shown in Figure 8a. The distortion by the interferer at different angles could not be removed by applying range-FFT and the monitored respiration rate was not accurate.

Applying a beamforming could suppress the distortion by the interferer. Figure 13c shows the phase signal applying range-FFT and beamforming. Comparing the phase signals not applying beamforming in Figure 13b and applying beamforming in Figure 13c, the distorted part of the signal was recovered. In the time domain, the improvement can be seen by comparing the signals in Figure 13a,c. Applying FFT to the signals in Figure 13a,c, we calculated the amount of improved SINR. The amount of improved SINR was almost 22 dB applying the range-FFT and BF. Then, we could calculate the respiration rate. The average respiration rate was 17 BPM and the monitored average value of reference sensor was 17 BPM.

From the phase signal in Figure 13b, using our algorithm, extracted heartbeat waveform is shown in Figure 13c. The measured respiration and heartrates for 100 s were drawn in Figure 14. When there was interferer, the algorithm performances of the three cases (raw phase signal, applying range-FFT, and applying range-FFT and beamforming) were compared. Additionally, in only applying range-FFT and beamforming, the respiration and heartrates were sustained. Because the noise by the interferer’s movements were added, the fluctuation of heartrate frequently occurred for the case of applying range-FFT.

### 5.3. Results of Accuracy for Five People

For five people, we measured the respiration and heartbeat rates. The results are listed in Table 3, Table 4, Table 5 and Table 6. Regardless of the presence of an interferer, the respiration and heartrates monitored by the Doppler radar were inaccurate and not constant even though the error values are listed in Table 3 and Table 5. The results monitored by FMCW radar with beamforming are listed in Table 4 and Table 6 for the cases with and without an interferer, respectively. 

The accuracy was calculated by averaging the percentage value of difference between results measured by our radar and reference sensor [43]. The equation is shown in Equation (4).
(4)$$Accuracy\left(\%\right)=\frac{1}{d}\sum_{i=0}^{d}\left(100-\frac{\left|R_{radar}\left(t\right)-R_{ref}\left(t\right)\right|}{R_{ref}\left(t\right)}\cdot 100\right)$$

The $R_{radar}\left(t\right)$ and $R_{ref}\left(t\right)$ are respiration or heartrates measured by our radar and reference sensor, respectively.

The averaged accuracy of respiration and heartbeat rates for five people were more than 90%. The error was mainly caused by the difference in length of time window calculating the respiration and heartrates. 

## 6. Conclusions

In this study, we proposed a method for improving the accuracy of non-contact monitoring of a patient’s respiration and heartrates when the patient is lying on the surgical bed and the surgeon as interferer stands adjacent to it. In this application, the radar should be attached on back of the bed. Because the radar was installed on the back of the bed, the radar signal was noisy due to multiple reflections. Using a 60 GHz FMCW radar system with beamforming we could reduce the noise and extract respiration and heartrate precisely. More than 90% accuracy was achieved even though the surgeon as interferer stood next to patient.

The FMCW radar IF signal model, including the respiration and heartbeat, was proposed to analyze the radar signal process. The respiration waveform could be modeled by sinusoidal function in the phase signal. However, because the movement of heartbeat is not sinusoidal wave, heartbeat waveform was modeled by repeated wavelet function. Using the proposed model, the experimental results were analyzed. 

There are some future works to improve our system. First, the algorithm tracking the target will be applied for removing the effect of the patient’s supine position. Second, the more advanced algorithm will be applied, finding heartbeat from raw phase signal such as empirical mode decomposition (EMD) and wavelet transform.

## Figures and Tables

**Figure 1 sensors-22-08167-f001:**
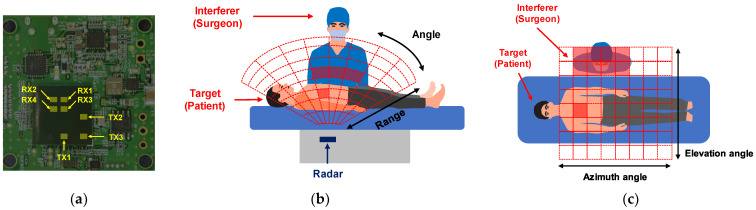
Scenario for monitoring vital signs of the patient lying on the surgical bed. (**a**) The used radar module; (**b**) is side–view; (**c**) and top–view.

**Figure 2 sensors-22-08167-f002:**
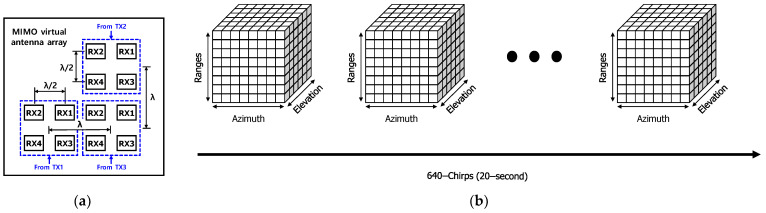
(**a**) The 12-virtual array RX channels generated by 3-TX and 4-sRX antennas and; (**b**) the radar cube generated by 12-virtual array RX channels.

**Figure 3 sensors-22-08167-f003:**
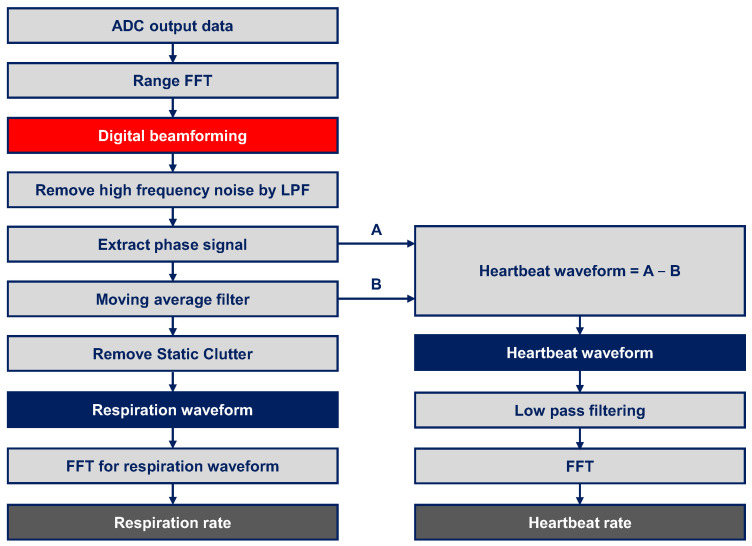
The flow chart of signal-processing algorithm monitoring respiration and heartbeat rates.

**Figure 4 sensors-22-08167-f004:**
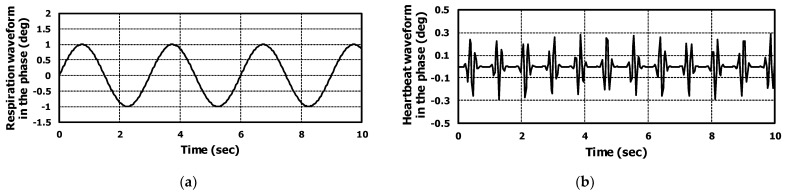
The simulation results of (**a**) respiration and (**b**) heartbeat signal model. The target information of respiration and heartbeat rates was 70 and 20, respectively.

**Figure 5 sensors-22-08167-f005:**
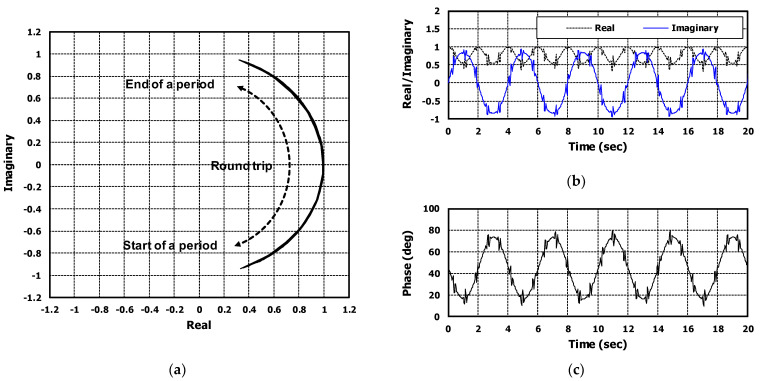
The signals simulated by proposed radar IF signal model including respiration and heartbeat signal. The signals were shown in (**a**) IQ domain and (**b**) time domain. (**c**) is the extracted phase from real and imaginary signals.

**Figure 6 sensors-22-08167-f006:**
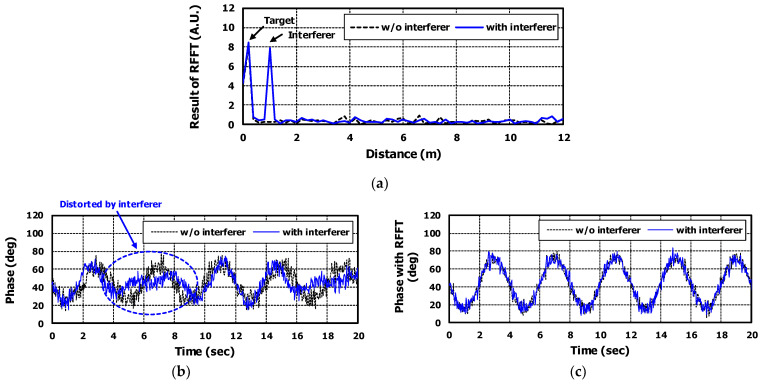
The simulation results for target information based on Table 1. The black dashed lines are results of the case without interferer and the blue solid lines are results of the case with interferer. (**a**) is the distance information of target and interferer obtained by range–FFT. (**b**) is the phase signal without applying range–FFT and (**c**) is the phase signal with applying range–FFT.

**Figure 7 sensors-22-08167-f007:**
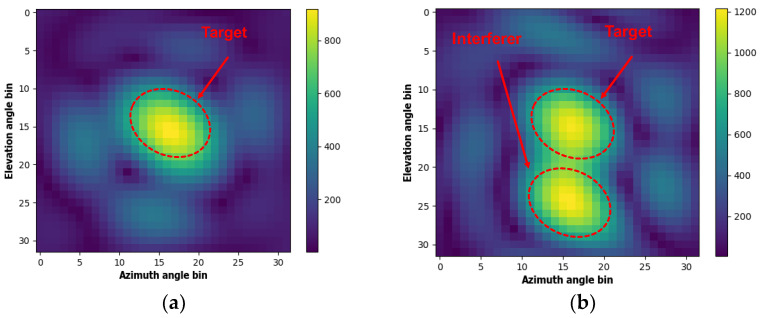
The heat map obtained by beamforming (**a**) without interferer and (**b**) with interferer.

**Figure 8 sensors-22-08167-f008:**
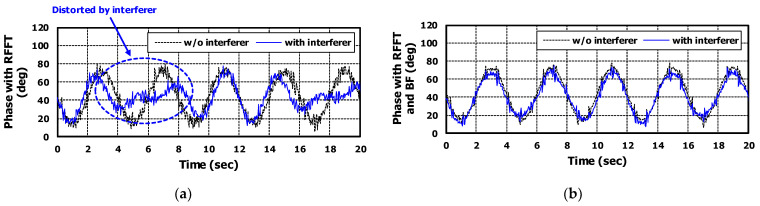
The simulation results for target information based on Table 2. The black dashed lines are results of the case without interferer and the blue solid lines are results of the case with interferer. (**a**) is the phase signal with applying range-FFT and (**b**) is the phase signal with applying range-FFT and beamforming.

**Figure 9 sensors-22-08167-f009:**
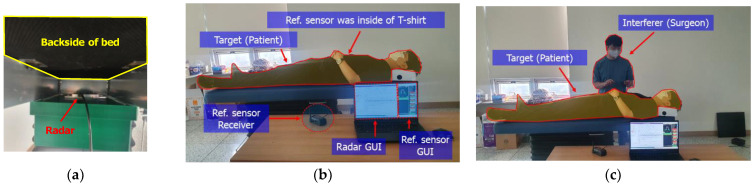
The photographs of experiment setup in laboratory. (**a**) is the photo of radar module installed on back of surgical bed, (**b**) is the photo of monitoring patient’s vital signs, and (**c**) is the photo of monitoring patient’s vital signs with surgeon as interferer.

**Figure 10 sensors-22-08167-f010:**
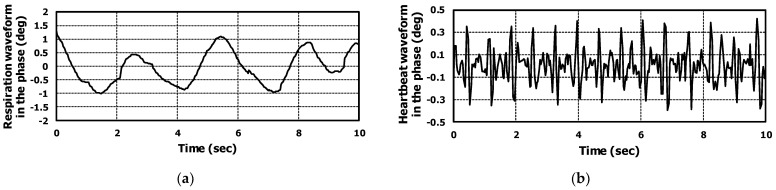
The extracted waveform of (**a**) respiration and (**b**) heartbeat from measured raw data of radar.

**Figure 11 sensors-22-08167-f011:**
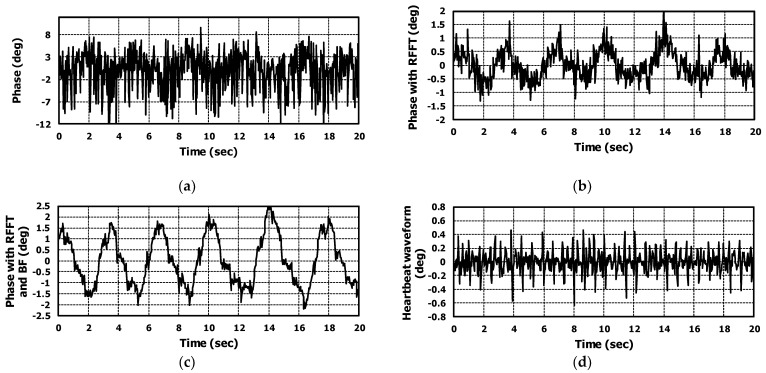
For the case without interferer, the extracted phase signal from measured raw data of radar. (**a**) is the phase signal not applying range-FFT and beamforming, (**b**) is the phase signal applying ranges-FFT, (**c**) is the phase signal applying range-FFT and beamforming, and (**d**) is the extracted heartbeat waveform from the (**c**).

**Figure 12 sensors-22-08167-f012:**
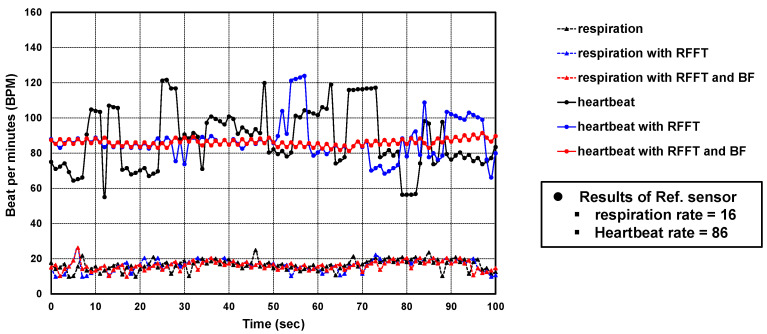
Comparison of algorithm performance without interferer.

**Figure 13 sensors-22-08167-f013:**
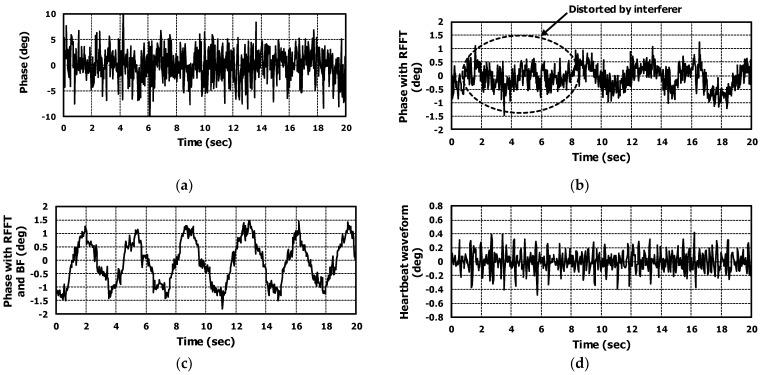
For the case with interferer, the extracted phase signal from measured raw data of radar. (**a**) is raw phase signal, (**b**) is the phase signal applying range-FFT, (**c**) is the phase signal applying range-FFT and beamforming, and (**d**) is the extracted heartbeat waveform from the (**c**).

**Figure 14 sensors-22-08167-f014:**
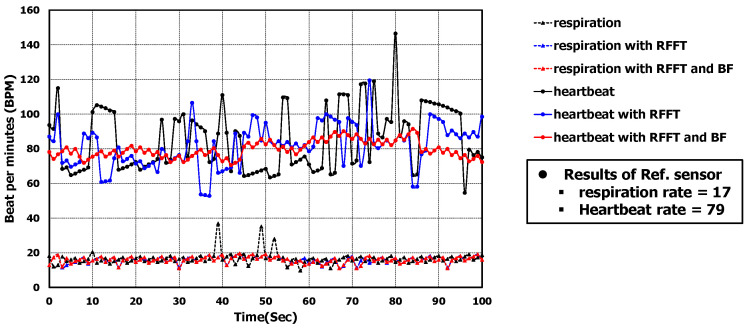
Comparison of algorithm performance with interferer.

**Table 1 sensors-22-08167-t001:** Simulation information with interferer in a different range.

	Target (Patient)	Interferer (Surgeon)
Respiration rate (BPM)	15	20
Heartbeat rate (BPM)	80	70
Distance (cm)	20	100
Azimuth angle (deg)	0	0
Elevation angle (deg)	0	0

**Table 2 sensors-22-08167-t002:** Simulation information with interferer in a different angle.

	Target (Patient)	Interferer (Surgeon)
Respiration rate (BPM)	15	20
Heartbeat rate (BPM)	80	70
Distance (cm)	20	20
Azimuth angle (deg)	0	30
Elevation angle (deg)	0	0

**Table 3 sensors-22-08167-t003:** The measurement accuracy of respiration and heartrates monitored by the Doppler radar for the case without interferer.

No.	Reference Sensor		Radar (Doppler)		Error (MAPE)		Error (MAE)		Error (RMSE)		Accuracy	
	RR (BPM)	HR (BPM)	RR (BPM)	HR (BPM)	RR (%)	HR (%)	RR (BPM)	HR (BPM)	RR (BPM)	HR (BPM)	RR (%)	HR (%)
1	9.74	73.42	17.67	76.00	83.18	15.51	8.11	11.33	9.84	14.39	16.82	84.70
2	19.52	64.84	19.27	76.94	12.36	22.19	2.41	14.43	4.39	20.84	87.64	80.36
3	18.62	75.27	18.08	82.07	10.42	19.24	1.94	14.30	2.59	18.21	89.58	82.78
4	17.64	64.41	18.22	79.80	14.04	28.35	2.45	18.14	3.15	22.09	85.96	75.88
5	16.40	75.69	18.84	79.40	21.68	16.68	3.56	12.57	5.18	15.28	78.32	83.85

**Table 4 sensors-22-08167-t004:** The measurement accuracy of respiration and heartrates monitored by the FMCW radar with beamforming for the case without interferer.

No.	Reference Sensor		Radar (Doppler)		Error (MAPE)		Error (MAE)		Error (RMSE)		Accuracy	
	RR (BPM)	HR (BPM)	RR (BPM)	HR (BPM)	RR (%)	HR (%)	RR (BPM)	HR (BPM)	RR (BPM)	HR (BPM)	RR (%)	HR (%)
1	9.74	73.42	10.28	75.11	9.26	3.93	0.89	2.88	1.07	3.29	90.74	96.07
2	19.52	64.84	19.67	70.01	3.35	7.87	0.65	5.17	0.84	5.40	96.65	92.00
3	18.62	75.27	19.34	78.41	6.9	5.15	1.32	3.81	1.65	4.55	93.10	94.84
4	17.64	64.41	19.02	67.28	9.69	5.56	1.71	3.50	2.03	4.53	90.31	94.44
5	16.40	75.69	17.47	77.62	7.76	4.85	1.27	3.60	1.50	4.32	92.24	95.15

**Table 5 sensors-22-08167-t005:** The measurement accuracy of respiration and heartrates monitored by the Doppler radar for the case with interferer.

No.	Reference Sensor		Radar (Doppler)		Error (MAPE)		Error (MAE)		Error (RMSE)		Accuracy	
	RR (BPM)	HR (BPM)	RR (BPM)	HR (BPM)	RR (%)	HR (%)	RR (BPM)	HR (BPM)	RR (BPM)	HR (BPM)	RR (%)	HR (%)
1	10.00	71.53	19.59	81.38	95.89	20	9.59	14.27	11.21	18.13	4.11	81.35
2	17.78	64.62	16.98	77.65	19.79	26.77	3.51	17.23	4.40	20.74	80.21	75.20
3	16.02	75.07	17.68	82.98	21.15	20.30	3.61	15.31	4.88	19.37	78.85	82.08
4	19.85	54.29	24.2	89.05	30.74	64.93	6.09	35.16	7.29	38.00	69.26	43.64
5	15.36	73.96	19.42	82.42	24.07	51.91	4.83	28.21	6.26	31.37	63.46	82.04

**Table 6 sensors-22-08167-t006:** The measurement accuracy of respiration and heartrates monitored by the FMCW radar with beamforming for the case with interferer.

No.	Reference Sensor		Radar (Doppler)		Error (MAPE)		Error (MAE)		Error (RMSE)		Accuracy	
	RR (BPM)	HR (BPM)	RR (BPM)	HR (BPM)	RR (%)	HR (%)	RR (BPM)	HR (BPM)	RR (BPM)	HR (BPM)	RR (%)	HR (%)
1	10.00	71.53	9.89	73.42	9.78	3.48	1.00	2.46	1.23	3.00	90.04	96.52
2	17.78	64.62	18.31	66.05	4.33	3.20	0.75	2.10	0.85	2.53	95.78	96.75
3	16.02	75.07	18.58	78.9	9.82	5.76	1.66	4.25	1.91	4.86	90.18	94.24
4	19.85	54.29	20.81	57.38	8.94	5.40	1.52	3.09	1.81	3.28	92.34	94.28
5	15.36	73.96	16.15	75.83	7.84	6.09	1.19	4.65	1.43	5.44	92.16	93.70

## Data Availability

Not applicable.

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
