# Peer review of "Non-Contact VITAL Signs Monitoring of a Patient Lying on Surgical Bed Using Beamforming FMCW Radar"

_sensors, 2022, doi:10.3390/s22218167_

Round 1

Reviewer 1 Report

In this study, the authors proposed a method for improving the accuracy of patients' respiration and heart rates using non-contact sensors.

The paper is well written. However, I have a few suggestions that should be considered before publication:

1) In the introduction section, please add/elaborate on the description of the usage of radar in the healthcare industry in general as well.

2) Authors must include the beam-width angle of the radar used in this study in the section. 

3) How was the clutter removed? Please include the steps mentioned here: (at vital signs extraction algorithm)

Effects of Receiver Beamforming for Vital Sign Measurements Using FMCW Radar at Various Distances and Angles by Shahzad Ahmed , Junbyung Park and Sung Ho Cho.

4) at the end of the introduction, please categorically mention the novelty of this paper.

5) why the authors calculated only MAPE? would be nice to see the results with MAE or RMSE? 

Author Response

Thank you for your kind comment.

I agree your comment and modified my paper.

1) In the introduction section, please add/elaborate on the description of the usage of radar in the healthcare industry in general as well.

Answer:

I added the usage of radar in the healthcare, automotive industry, and disaster situation.

2) Authors must include the beam-width angle of the radar used in this study in the section. 

Answer:

The beamwidth of the patch antenna in our radar was 120degree. The angular resolution was 29degree using 4X4 MIMO architecture which is shown in Fig. 2(a).     

3) How was the clutter removed? Please include the steps mentioned here: (at vital signs extraction algorithm)

Answer:

Thank you for your comment. I omitted the clutter remove in algorithm flow chart. For the respiration, I modified the algorithm flow chart, and then inserted the context of removing clutter. However, for heartbeat, the clutter was removed by subtracting raw phase signal from respiration waveform.

Effects of Receiver Beamforming for Vital Sign Measurements Using FMCW Radar at Various Distances and Angles by Shahzad Ahmed , Junbyung Park and Sung Ho Cho.

4) at the end of the introduction, please categorically mention the novelty of this paper.

Answer:

Thank to your comment, the introduction was improved. I described the novelty at the end of the introduction.

5) why the authors calculated only MAPE? would be nice to see the results with MAE or RMSE? 

Answer:

I used the MAPE to show the percentage. However, I added the MAE and RMSE.

Reviewer 2 Report

The paper concerns an improvement of radar techniques for monitoring the respiration and heart rate of a patient lying on a surgical bed, taking also into account the interference due to the surgeon's presence. This improvement is mostly based on employing a frequency-modulated continuous wave (FMCW) radar system and on beamforming.

I have found the paper well written and organized; the subject is interesting and well developed. Therefore, I recommend it for publication.

Author Response

Thank you for your kind comment.

Reviewer 3 Report

1. The combination of FMCW and beam forming have widely used in many works to enhance the performance of radar. Authors should highlight the novelty of this work.

2.  In the abstract, authors claim that the FMCW radar with beamforming improved almost 20 dB of SINR for vital signal when comparing a CW Doppler radar.  However, the measurement of 20 dB SINR improment can not be found in the paper.

3. The last paragraph on page 5 discribes that the Fig. 6(c) shows the SINR was improved by 9.08 dB. How to calculate the 9.08 dB improvement  from Fig. 6(c)?

4. The error of respiration rate and heart rate without using FMCW and beam forming should be added in Tables 3 and 4 to demonstrate the improvement of performance.

Author Response

Thank you for your comment.

I agree your comment and modified my paper.

1. The combination of FMCW and beam forming have widely used in many works to enhance the performance of radar. Authors should highlight the novelty of this work.

Answer:

  1. We try to apply vital-sign monitoring radar to surgical bed environment. Installing the radar on the back of bed make additional noise which generated by electromagnetic-wave reflections from the bed structures. Almost of vital-sign monitoring radar was set at the front of target therefore it was not suffered by reflections because space between radar and target was opened.
  2. The respiration and heartbeat signal were modeled in radar IF signal.

2. In the abstract, authors claim that the FMCW radar with beamforming improved almost 20 dB of SINR for vital signal when comparing a CW Doppler radar.  However, the measurement of 20 dB SINR improment can not be found in the paper.

Answer:

Thank you for your comment. I had a mistake. The 20dB of improvement mean the measurement without interferer. Main component of the improvement was effect of multiple reflection of wave due to environment around radar such as steel frame. Using beamforming, noise from different angles due to multiple reflection can be reduced. Because there was not interferer in this case, the calculation results is SNR(signal-to-noise-ratio). I attached the value of improved SINR for the case with interferer.

3. The last paragraph on page 5 discribes that the Fig. 6(c) shows the SINR was improved by 9.08 dB. How to calculate the 9.08 dB improvement from Fig. 6(c)?

Answer:

The SINR was calculated in frequency domain. The value of 9.08dB is difference between the SINRs of applying range-FFT and not applying range-FFT. The case of applying range-FFT is blue lines in Fig. 6(c). And the case of not applying range-FFT is blue lines in Fig. 6(b). The FFT was applied to the phase signals shown in Fig. 6(b) and (c). And then the ratio between signal power and sum of noise and interference power was calculated.

4. The error of respiration rate and heart rate without using FMCW and beam forming should be added in Tables 3 and 4 to demonstrate the improvement of performance.

Answer:

I attached the result of doppler radar in Table 3 and 5. And the result of the FMCW radar with beamforming were listed in Table 4 and 6. The result of doppler radar (=not using FMCW and beamforming) almost cannot monitor the respiration and heart rates. Like Fig. 12 and 14 the monitored values were not consistent. Especially, because the respiration signal of subject 1 was small, significant error of the result was achieved. I had a mistake in previous Tables. Even though the error values were not changed, the averaged respiration and heart rates of reference sensor and radar were switched.

Reviewer 4 Report

Summarizing the article, the key result is that angular resolution (beamforming) and distance resolution (range FFT) can reduce the influence of an interferer. This result is neither surprising nor novel. The article does not mention key parameters of the system (BW, antenna parameters) and the resulting spatial and angular resolution. The presented problem is similar to the problem of extracting the vital signs of multiple targets in sight of the radar system. This problem has been treated in detail in literature:
https://doi.org/10.1038/s41598-022-11671-1
https://doi.org/10.1109/JERM.2021.3082807
https://doi.org/10.1109/JERM.2022.3143431
https://doi.org/10.1186/s13634-021-00812-9

In my opinion, the surgery scenario is not a convincing application field for a vital sign radar with angular resolution. On the one hand, various contact-based sensors (oxymetry, blood pressure, body temperature) are already attached to the patient during surgery without distracting the surgeon. On the other hand, the biggest advantage of radar-based vital sign sensors, which is to allow the patient to move freely, is not relevant for an anesthetized patient.

The article needs to be revised fundamentally in terms of grammar, spelling and formal aspects.

Further comments regarding the content:

L40: "The second characteristic is that most of the radar systems monitoring vital-signs are implemented at frequencies under 10 GHz." - Is there any proof of this statement? From my experience, most vital sign radars have been implemented in the ISM bands at 24GHz or 61GHz.

L87: It would be beneficial for the reader to know the filter cutoff frequencies. Please include them in your article.

Figures 4/5: Please mention in the caption that the presented signals are simulations, not measurements.

L145: Real measured data contains a DC offset (both in Re and Imag), which has to be compensated before extracting the phase. Your simulated signal does not show a DC offset. Please comment on this.

Which bandwidth/ramp parameters were used by radar system? How about the simulations? This is very relevant since bandwidth is directly linked to distance resolution and therefore target separation.

Figure 12: Please include the measured values of the reference sensor as ground truth.

Table 3: How was the accuracy calculated? Pleas provide statistics here (precision, recall, F1).

Formal comments/typos:
- Please introduce a blank space between the last word of a sentence and the following reference(s).
- Articles are missing in many sentences
- Please add a (half) blank space between number and unit, i.e. "20dB" -> "20 dB"
L12: "movement" -> "movements"
L15: blank space before "(FFT)" missing
L18: blank space before "(CW)" missing
L20: "was existed" -> "was located"
L26: "vital-signs" was written without hyphen before
L53: "When considering of" - remove the "of"
L53: "is important variable" - is an important variable"
L67: blank space in "60GHz" is missing
L68: "which shown" -> "which is shown"
L78: "which architecture shown" -> "which architecture is shown"
L101: "should be separate" -> "should separate"
L118, 138: "complex plain" -> "complex plane"
L156: "session" -> "section"
L163: "was shown" -> "is shown"
Table 1, 4: There is a page break in between the table, which separates column headings and the columns. Please make sure to have the entire table positioned on one page.
L202: "Beamforming" -> "beamforming"
L222: remove the underscore
L223: "was wear on" -> "was wearing"
L239 "seismocardiogram(SCG)" -> "seismocardiogram (SCG)"

Author Response

Summarizing the article, the key result is that angular resolution (beamforming) and distance resolution (range FFT) can reduce the influence of an interferer. This result is neither surprising nor novel. The article does not mention key parameters of the system (BW, antenna parameters) and the resulting spatial and angular resolution. The presented problem is similar to the problem of extracting the vital signs of multiple targets in sight of the radar system. This problem has been treated in detail in literature:

Answer:

We try to apply vital-sign monitoring radar to surgical bed environment. Installing the radar on the back of bed make additional noise which generated by electromagnetic-wave reflections from the bed structures. Almost of vital-sign monitoring radar was set at the front of target therefore it was not suffered by reflections because space between radar and target was opened. 

In my opinion, the surgery scenario is not a convincing application field for a vital sign radar with angular resolution. On the one hand, various contact-based sensors (oxymetry, blood pressure, body temperature) are already attached to the patient during surgery without distracting the surgeon. On the other hand, the biggest advantage of radar-based vital sign sensors, which is to allow the patient to move freely, is not relevant for an anesthetized patient.

Answer:

I agree your opinion and I modified the motivation of non-contact monitoring. 

L40: "The second characteristic is that most of the radar systems monitoring vital-signs are implemented at frequencies under 10 GHz." - Is there any proof of this statement? From my experience, most vital sign radars have been implemented in the ISM bands at 24GHz or 61GHz.

Answer:

I agree your opinion and I modified.

L87: It would be beneficial for the reader to know the filter cutoff frequencies. Please include them in your article.

Answer:

I include the cut-off frequency.

Figures 4/5: Please mention in the caption that the presented signals are simulations, not measurements.

Answer:

I agree your comment and I modified the captions of Fig. 4 and 5.

L145: Real measured data contains a DC offset (both in Re and Imag), which has to be compensated before extracting the phase. Your simulated signal does not show a DC offset. Please comment on this.

Answer:

Your comment is correct. Because the signal model is consisting of sine and cosine functions, I can add the I/Q DC offset. However, because to conceptually explain the effect of interferer, I didn’t add the I/Q DC offset.

Which bandwidth/ramp parameters were used by radar system? How about the simulations? This is very relevant since bandwidth is directly linked to distance resolution and therefore target separation.

Answer:

The bandwidth(=750MHz), ramp parameter(chirp slope=24) are same for measurement and simulation. And then the range resolution was 20 cm which calculated by below equation.

Figure 12: Please include the measured values of the reference sensor as ground truth.
Answer:

I attached the respiration and heart rate values measured by reference sensor.

Table 3: How was the accuracy calculated? Pleas provide statistics here (precision, recall, F1).
Answer:

The respiration and heart rate are continuous values and can not judge pass or fail. So, I hesitate the accuracy calculation. A paper shows the accuracy of respiration using below equation. [Lee, J.M.; Song, H.; Shin, H.C. Respiration Rate Extraction of Moving Subject Using Velocity Change in FMCW Radar. Ap-plied Sciences 2022, Vol. 12, Page 4114 2022, 12, 4114, doi:10.3390/APP12094114].

I also calculated the accuracy this equation which is in Eq (4). The  and  are respiration or heart rates measured by our radar and reference sensor, respectively. 

Formal comments/typos:

Answer:

Thank you for your kind condsideration. I fixed.

Round 2

Reviewer 1 Report

The authors have fixed my concerns, I am happy to accept it.

Reviewer 3 Report

All my comments have been properly addressed in the revised manuscript. No further suggestions.

Reviewer 4 Report

Thank you for including my review comments.